# Pharmacological Effects and Potential Clinical Usefulness of Polyphenols in Benign Prostatic Hyperplasia

**DOI:** 10.3390/molecules26020450

**Published:** 2021-01-16

**Authors:** Kensuke Mitsunari, Yasuyoshi Miyata, Tomohiro Matsuo, Yuta Mukae, Asato Otsubo, Junki Harada, Tsubasa Kondo, Tsuyoshi Matsuda, Kojiro Ohba, Hideki Sakai

**Affiliations:** Department of Urology, Nagasaki University Graduate School of Biomedical Sciences, Nagasaki 852-8501, Japan; ken.mitsunari@gmail.com (K.M.); tomozo1228@hotmail.com (T.M.); ytmk_n2@yahoo.co.jp (Y.M.); a.06131dpsc@gmail.com (A.O.); harada_junki_19881027@yahoo.co.jp (J.H.); t-udonko@nagasaki-u.ac.jp (T.K.); matsudatsuyoshi9251@gmail.com (T.M.); ohba-k@nagasaki-u.ac.jp (K.O.); hsakai@nagasaki-u.ac.jp (H.S.)

**Keywords:** polyphenols, flavonoids, pharmacological effect, benign prostatic hyperplasia

## Abstract

Benign prostatic hyperplasia (BPH) is arguably the most common benign disease among men. This disease is often associated with lower urinary tract symptoms (LUTS) in men and significantly decreases the quality of life. Polyphenol consumption reportedly plays an important role in the prevention of many diseases, including BPH. In recent years, in addition to disease prevention, many studies have reported the efficacy and safety of polyphenol treatment against various pathological conditions in vivo and in vitro. Furthermore, numerous studies have also revealed the molecular mechanisms of the antioxidant and anti-inflammatory effects of polyphenols. We believe that an improved understanding of the detailed pharmacological roles of polyphenol-induced activities at a molecular level is important for the prevention and treatment of BPH. Polyphenols are composed of many members, and their biological roles differ. In this review, we first provide information regarding the pathological roles of oxidative stress and inflammation in BPH. Next, the antioxidant and anti-inflammatory effects of polyphenols, including those of flavonoids and non-flavonoids, are discussed. Finally, we talk about the results and limitations of previous clinical trials that have used polyphenols in BPH, with particular focus on their molecular mechanisms of action.

## 1. Introduction

Benign prostatic hyperplasia (BPH) is a health concern that is expected to increase with age, and its incidence increases with life expectancy [1]. BPH is a non-malignant enlargement of the prostate and can cause obstructive and irritating lower urinary tract symptoms (LUTS); this proliferative change affects smooth muscle and epithelial cells within the transition zone of the prostate [2]. Ultimately, these changes cause the compression of the urethra as a bladder outlet obstruction (BOO), manifested as LUTS [3]. Typical symptoms include pollakiuria, nocturia, urgency, decreased urinary dysfunction, dysuria, BOO, and residual urine [4]. BPH is observed in older men, and its incidence depends on age. BPH usually begins in men aged approximately 40–45 years and causes few symptoms in the early stages. Symptoms develop with age, and more than half of all men develop BPH [5]. These symptoms are known as LUTS [6].

The molecular and interstitial mechanisms associated with this etiology are not fully understood, but several factors, including inflammatory mediators, hormones, dietary factors, and environmental and oxidative stress, may be involved [7]. An increase in androgen dihydrotestosterone (DHT) is thought to stimulate prostate cell growth. Furthermore, the association between obesity, hypertension, hyperglycemia, dyslipidemia, and BPH has also been reported [8]. Insulin resistance induces hyperinsulinemia, which can lead to sympathetic hypertonia, increased prostatic smooth muscle proliferation and tone, and, finally, an appearance of LUTS [9]. According to a meta-analysis, people with metabolic syndrome display significantly larger prostate and transition area volumes of 1.8 and 3.67 mL, respectively, than those without any metabolic syndrome [10].

Prostate inflammation also plays an important role in hyperplasia [11,12]. A clinical study of 5α-reductase inhibitors reported that patients with inflammatory findings on prostate biopsy specimens had higher prostate volumes, had higher symptom scores [12], and were also at higher risk of urinary retention in inflamed cases [11]. Cytokines derived from inflammatory cells also induce growth factors. Among them, interleukin (IL)-1α induces fibroblast growth factor (FGF)-7, and the IL-1α-related system is involved in proliferation [13]. It is speculated that chronic inflammation and increased immune response brought about by bacterial infections or other foreign antigens cause the remodeling of the prostate structure, resulting in an enlargement of the prostate [14].

All things considered, the etiology of BPH is complex and has many uncertainties. As a result, treatment can often be multi-targeted. In this regard, phytotherapy, which is believed to have pharmacological properties, such as anti-androgens, anti-proliferation, antioxidant, and anti-inflammatory properties, may be useful. Isoflavonoids and lignans, which are abundant in vegetables, grains, and soybeans, are presumed to function as BPH suppressors. These exert some degree of estrogenic effects and are thought to inhibit prostatic cell growth [15]. These pharmacological products also exhibit 5α-reductase inhibitory activity and angiogenesis inhibitory activity [16]. Thus, phytotherapy may be therapeutic in the prevention of mild-to-moderate prostate disease. In recent years, phytochemicals found in fruits, vegetables, and tea have gained the interest of researchers because they are less toxic, more effective, and economical.

The structures and biological activities of polyphenols have already been discussed in numerous previous reports. Additionally, data regarding the clinical significance and pharmacological roles of polyphenols in BPH have also been discussed in previously published reviews [17,18]. However, few studies have focused on the molecular mechanisms of the pharmacological effects of polyphenols in BPH. The pathological characteristics and growth steps of BPH are strongly modulated by oxidative stress, inflammation, and angiogenesis, as discussed in followed sections. Importantly, these pathological activity-related molecules are also targets of the pharmacological effects of polyphenols in various pathological conditions, including BPH. Therefore, in this review, we introduce the relationships between oxidative stress, chronic inflammation, and/or angiogenesis, BPH, and polyphenols for the first time. Furthermore, we discuss the in vivo and in vitro pharmacological effects and molecular mechanisms of the anti-growth function of polyphenols in BPH.

## 2. A Brief Overview of Inflammation and Oxidative Stress in BPH

### 2.1. Pathological Roles of Inflammation and Oxidative Stress

Inflammation is an important protective response against tissue damage or pathogens; however, uncontrolled chronic inflammatory reactions can cause various chronic diseases, including kidney disease, diabetes mellitus, and malignancies [19,20,21]. In addition, inflammation is associated with many diseases such as microbial and viral infections, exposure to allergens, radiation and toxic chemicals, autoimmune diseases, obesity, alcohol consumption, and tobacco use [22]. Inflammation is initiated by the synthesis and secretion of inflammatory cytokines, such as tumor necrosis factor-α (TNF-α), IL-1β, IL-6, IL-12, and interferon (IFN)-γ, and by macrophages in response to inflammatory injury [23]. It has also been reported that various inflammatory stimuli, such as excess reactive oxygen species (ROS) produced during oxidative metabolism, initiate the inflammatory process, leading to the synthesis and secretion of pro-inflammatory cytokines [22]. Inflammatory cytokines and their receptors promote mitogen-activated protein kinase activity, resulting in the activation of transcription factors NF-κB and activator protein (AP)-1 [24]. These transcription factors amplify the inflammatory response by activating the expression of various genes, such as cytokines, chemokines, adhesion molecules, inducible nitric oxide synthase (iNOS), and cyclooxygenase-2 (COX-2) [25]. The mechanism by which oxidative stress causes inflammation is unknown, but oxidative stress is clearly associated with many chronic inflammatory diseases and is especially important. Oxidative stress occurs when there is an imbalance between antioxidants and pro-oxidants, which in turn favors oxidation and can damage DNA, proteins, and lipids [26]. The imbalance between production and detoxification of ROS/free radicals can induce tissue damage [27]. Most chronic diseases that occur with increased ROS production cause oxidative stress and the oxidation of various proteins [28]. Protein oxidation induces the release of inflammatory signals [29]. Increased ROS production leads to inadequate antioxidant defense mechanisms; the disruption of proteins, lipids, and DNA; the disruption of cell function and cell death; and the induction of oxidative stress [30,31].

### 2.2. Benign Prostate Hyperplasia and Inflammation

Several investigators have suggested that inflammation plays important roles in the pathogenesis of prostate diseases, including BPH and prostate cancer [21,32]. Inflammation affects the balance between prostate cell proliferation and apoptosis by increasing factors such as cytokines, COX-2, and oxidative stress around the prostate [32]. These factors stimulate proliferation and minimize apoptosis [32]. In a study involving the etiology of BPH, the differential expression of cytokines and growth factors in BPH tissue suggested the role of inflammation in BPH development [33]. Various clinical trials have also suggested the role of inflammation in BPH. Previous studies have shown that the degree of BPH inflammation correlates with prostate size. A statistically significant correlation between chronic inflammation and International Prostate Symptom Score (IPSS) was observed, with severe inflammation associated with higher IPSS scores [12]. It has been reported that 57% of patients with prostatitis have a history of BPH and are at a significantly higher risk of developing hypertrophy and acute urinary retention [34]. Furthermore, it was shown that the use of COX-2 inhibitors in combination with 5α-reductase inhibitors could increase the apoptosis index in BPH tissues [34]. Pathological data were analyzed from 374 patients who underwent transurethral resection of the prostate (TURP) for BPH, and urinary retention with findings of acute and/or chronic prostatic inflammation were observed in 70% of the patients [11]. In another clinical study, the pathological data were analyzed from 3942 patients who underwent surgical treatment for BPH, and 43.1% primarily displayed chronic inflammation [35]. 

Inflammation distribution changed significantly with prostate volume, with a statistically significant correlation between chronic inflammation and prostate volume [35]. Inflammation is one of the factors that produces free radicals/ROS in the prostate and is associated with oxidative stress [32]. iNOS is induced and expressed in inflammatory cells, and it contributes to the pathogenesis of inflammation. iNOS is not detected in the normal prostate, but it is reported to be expressed in the prostate of all BPH patients [27]. COX-2 is associated with inflammation in BPH. Prostaglandins are synthesized from arachidonic acid via cyclooxygenase (COX-1 and COX-2) [36]. Prostaglandins, a group of inflammatory mediators, have been observed in the BPH tissues [37]. In addition, it has been reported that COX-2 is upregulated in the basal epithelial cells of BPH [36].

### 2.3. Benign Prostate Hyperplasia and Oxidative Stress

Oxidative stress is thought to be one of the mechanisms that triggers a series of reactions involved in the development and progression of BPH [31]. It has been reported that antioxidant levels are significantly reduced in prostate tissue during BPH [31]. In addition, human prostate tissue is susceptible to oxidative DNA damage due to the fast cell turnover and low levels of DNA repair enzymes. Therefore, BPH may be strongly associated with oxidative stress [27]. The antioxidant activities of glutathione (GSH), superoxide dismutase (SOD), glutathione peroxidase (GPx), and catalase were all significantly reduced in the prostate of an untreated BPH rat model [38]. A significant increase in prostatic lipid peroxidation was also observed. After treatment with finasteride or kolaviron, the parameters of these antioxidants improved significantly [39].

Thus, inflammation and oxidative stress are among the major pathological mechanisms underlying the development of BPH. In addition, we should note the facts that there are cross-talks reported between inflammation and oxidative stress. A schema of pathological mechanisms leading to BPH development via the regulation of inflammation and oxidative stress is shown in Figure 1. 

## 3. A Brief Overview of Polyphenols in Inflammation and Oxidative Stress

In recent years, natural polyphenols have been attracting attention. Polyphenols are the most abundant antioxidants in many foods such as fruits, vegetables, seeds, nuts, chocolate, wine, coffee, and tea [40,41,42]. More than 8000 polyphenols have been described to be present in foods and have been shown to play an important role in human health and its maintenance [43]. Various molecules, especially those involved in inflammation- and oxidative stress-related pathways, have been reported to play crucial roles in regulating the biological effects of polyphenols [44]. For example, regarding inflammation, although the activation of NF-κB is caused by chronic diseases via the initiation of the inflammatory process, polyphenols regulate the NF-κB-related pathway and act as anti-inflammatory factors by suppressing the production of inflammatory-related cytokines and chemokines [45,46]. Conversely, regarding oxidative stress, although the overproduction of ROS leads to tissue damage and initiates the process of inflammation [47], polyphenols can interact with ROS/reactive nitrogen species (RNS) and terminate the chain reaction before the cells are seriously affected [48]. Furthermore, other investigators have shown that the accumulation of free radicals and ROS in the body leads to oxidative stress, while polyphenols reportedly function as potent free radical scavengers and removers of ROS [49,50]. Thus, there is a general agreement that polyphenols are strong anti-inflammatory and antioxidant factors via complex mechanisms.

Dietary polyphenols are classified as either flavonoids or non-flavonoids, according to the number of phenol rings and the structural elements attached to these rings [51]. Non-flavonoid compounds include phenolic acids (hydroxybenzoic acid/hydroxycinnamic acid), tannins, lignans, stilbenes, and other phenolic compounds (curcumin and gingerol) [51]. The radical scavenging activity of polyphenols is associated with the substitution of hydroxyl groups in the aromatic rings of phenols [52]. The total phenolic content and total antioxidant activity of various fruit phytochemical extracts may be directly related. Fruits with a high total phenolic content have stronger antioxidant activity [53]. The most common dietary polyphenols are the flavonoid and non-flavonoid phenolic acids. Polyphenol-driven disease prevention is mainly achieved by antioxidant activity, and the above-mentioned types represent major antioxidant phytochemicals that contribute most to the antioxidant properties.

## 4. Management of Benign Prostate Hyperplasia with Polyphenols

The bioavailability of dietary polyphenols has not yet been fully elucidated, but there are reports that the prostate can easily take up polyphenols [54]. For example, polyphenols were detected in many tissue samples obtained from mice and rats using HPLC, one of which was prostate tissue, thus suggesting the bioavailability of polyphenols in prostate tissue [55]. These reports suggest that dietary polyphenols can exert their biological effects in the prostate. Treatments for BPH include conservative treatment with oral medication and surgical treatment, such as transurethral resection. Oral treatment with α1 blockers, phosphodiesterase 5 inhibitors, and 5α-reductase inhibitors are commonly used [56]. However, these demonstrate many side effects, such as hypotension, dizziness, gynecomastia, and sexual dysfunction, and they are not useful in all patients [57]. There are patients who experience a limited efficacy of these agents. Therefore, BPH treatment requires more efficient drugs with fewer side effects. Polyphenols are well-known extrinsic antioxidants, and here we introduce the effects of polyphenols on BPH. 

As mentioned above, dietary polyphenols are divided into two main compounds: flavonoids and non-flavonoids [51]. In this review, we discuss the pharmacological roles of polyphenols in BPH and their molecular mechanisms in two separate sections: flavonoids and others. 

### 4.1. Flavonoids

Flavonoids constitute the majority (up to 60%) of dietary polyphenols [58]. Flavonoids are known to be found in fruits, vegetables, legumes, red wine, and green tea. They are further subdivided into several subgroups, such as flavanols, isoflavones, and anthocyanidins according to their chemical structure. Herein, we introduce the most prominent flavonoids.

#### 4.1.1. Flavanols

Flavanols are known to be found in green and black tea [59]. Other significant levels of flavanols are found in a variety of edible plants, such as apples, grapes, cocoa, berries, plums, apricots, and nuts [60]. The general structure of flavanols is shown in Figure 2.

Unfortunately, there has been no report on the direct influence of flavanols on the etiology and development of BPH. However, several studies have investigated the roles of flavanol glycosides and flavanol-containing substances in BPH. For example, there was a report on the pharmacological effects of total flavanol glycosides from *Abacopteris penangiana* and its acid hydrolysate in testosterone-induced BPH rats [61]. This study showed that both the flavanol-related substances suppressed the growth of BPH via the downregulation of oxidative stress, inflammation, and cell survival [61]. Specifically, the oral intake of total flavanol glycosides and its acid hydrolysate increased antioxidative activities (SOD, GPx, and catalase (CAT)) and decreased oxidative stress (malondialdehyde (MDA)), tissue levels of inflammatory cytokines, and cell survival-related molecules (phospho-Akt, NF-κB, and B-cell lymphoma (Bcl)-2) [61]. 

In recent years, flavanols produced as secondary metabolites from *Tropaeolum tuberosum*, which is an indigenous plant used as a traditional medicine in South America, and *Tropaeolum tuberosum* extracts (tubers in particular) were found to exert antioxidative and anti-inflammatory effects [62]. Though the specific roles of flavanols in these substances were not elucidated, the authors suggested that *Tropaeolum tuberosum* extracts may be useful for treating BPH, and flavanols may play important roles in such biological and pharmacological activities [62]. 

#### 4.1.2. Isoflavones

Isoflavones are naturally occurring phytochemicals and are called phytoestrogens because they have an estrogen-like effect [63]. Isoflavones are found in soybeans and their processed products [64]. The main isoflavones in soybeans are genistein and daidzein [65]. Bioavailability primarily depends on the activity of the gut flora. Therefore, the absorption of isoflavones and their associated beneficial effects can vary considerably between individuals [66]. The general structure of isoflavones is shown in Figure 3.

Regarding the relationship between isoflavones and prostate diseases, research has reported that isoflavones exert a preventive effect on prostate carcinogenesis due to their phytoestrogen action [16]. In addition, many investigators have paid special attention to the anti-cancer effects of isoflavones in prostate cancer [67,68,69]. In contrast to prostate cancer, studies on the pharmacological effects of isoflavones on BPH are smaller. In this section, we introduce the preventive and anti-growth effects of isoflavones, such as genistein and daidzein, on BPH.

To our knowledge, anti-growth effects on human BPH tissues by genistein were first reported in 1998 [70], while other investigators further confirmed such genistein-induced growth inhibition on BPH [71]. In 2001, intake of a phytoestrogen-rich diet containing isoflavones was reported to decrease prostate weight in rats [72]. Interestingly, this study also showed that there were no significant differences in plasma levels of luteinizing hormone (LH) or the estradiol and testicular levels of testicular steroidogenic acute regulatory peptide between the phytoestrogen-rich included isoflavones and the phytoestrogen-free included isoflavones [72]. A study by Jarred et al. showed that the oral administration of red clover-derived isoflavones exerted inhibitory effects on prostatic growth in aromatase-knockout mice [73]. Based on this result, the authors speculated that isoflavones derived from red clover exerted anti-growth effects on BPH by acting as anti-androgenic agents [73]. However, another study showed that the increased expression of estrogen receptor (ER)-β and E-cadherin and a decrease in transforming growth factor (TGF)-β levels were associated with red clover-derived isoflavones in a mouse model [74]. In addition to red clover, a *Pueraria mirifica* extract contains daidzein and genistein, and it was found to decrease prostate weight in testosterone-induced BPH rats [75]. A previous study reported that red clover-derived isoflavones have suppressive effects on the smooth muscle contractions of isolated rat prostate glands, as well as antiproliferative effects [76]. This information is important for discussing the pharmacological effects of red clover-derived isoflavones in BPH. Importantly, such smooth muscle relaxant effects occurred at high concentrations, which is unlikely at doses used clinically [76]. 

A double-blinded, randomized control trial was conducted to clarify the clinical efficacy and safety of soy isoflavones in 176 BPH patients with LUTS [77]. In this study, the IPSS score, MOS Short-Form 36-Item Health Survey (SF-36) score, maximum urine flow rate, and residual urine volume were significantly improved in both the isoflavone intake (40 mg/day) group and the placebo group [77]. In comparison between the two groups, the maximum urine flow rate (*p* = 0.055), residual urine feeling (*p* = 0.05), and SF-36 (*p* = 0.02) showed improved tendencies in the isoflavone group [77]. Furthermore, this clinical trial showed that isoflavones are safe and well-tolerated [77]. In addition, a phase I/II study investigating the efficacy and safety of a combination of daidzein with isolase and zinc in 71 BPH patients with LUTS showed that combination therapy improved urinary tract conditions evaluated by IPSS, quality of life (QOL) questionnaires, and maximal urinary flow rate (Cmax), and adverse drug reactions and drug interactions were not found in any study population [78]. 

There is an opinion that isoflavone-containing supplements and foods are useful in the prevention of BPH and the management of patients at the watchful-waiting stage of BPH [79,80]. Based on previous reports, we agree with this opinion. However, there are various problems to be solved. For example, a previous study reported that although the prostatic concentration of daidzein in patients with BPH with prostate volume over 40 mL was similar to that in controls with a prostate volume <25 mL, the concentration of genistein in BPH patients was significantly (*p* = 0.032) lower than that in controls [81]. This result suggests the hypothesis that the clinical usefulness of isoflavone-containing substances may depend on the content of daidzein, genistein, and others. Furthermore, optimal dosage and administration methods are not clear in humans. First, in the discussion on the biological roles of isoflavones in prostate tissues, the most important characteristic is that of estrogen-like effects and the ability to inhibit 5 alpha-reductase activity [82,83]. Therefore, we must discuss the independent roles of these hormonal activities in order to understand the detailed biological and pharmacological roles of isoflavones. Thus, further studies are necessary to clarify the clinical significance and molecular mechanisms of isoflavone action on BPH, including pharmacokinetic studies, and to discuss prevention and treatment strategies for BPH.

#### 4.1.3. Anthocyanins

Anthocyanins are generally found in highly pigmented fruits [84]. More specifically, they are found in apples and berries, certain vegetables such as onions, tea, honey, wine, nuts, olive oil, cocoa, and grains [85]. They are the most prominent plant colorants, and the molecular structures of many anthocyanidins have been identified to date [85,86,87]. Anthocyanin is a water-soluble natural pigment that produces red, purple, and blue shades of various fruits, vegetables, grains, and flowers [86,87]. Its structure is shown in Figure 4

Regarding the pharmacological effects of anthocyanins in BPH, one study reported that black bean-derived anthocyanins suppressed prostate enlargement in BPH-induced rats [88]. In this study, BPH was induced by daily subcutaneous injections of testosterone propionate, and anthocyanins were orally administrated via drinking water to rats with BPH [88]. The mean/SD weights of prostate tissue at eight weeks in the control, BPH, and anthocyanin (40 mg/kg) groups were 674.17/28.24, 1098.33/131.31, and 323.00/22.41 mg, respectively, and statistical analyses showed that the mean prostate weight of the BPH group was significantly (*p* < 0.05) higher than that of the control, while the weight of the anthocyanin group was significantly lower (*p* < 0.05) than that of the control and BPH groups [88]. In addition, this study demonstrated that the frequency of apoptotic cells in the anthocyanin group was significantly higher than that in BPH group [88]. Though this study did not present any information regarding the molecular mechanisms of anthocyanin-induced apoptosis in BPH, another study showed that the downregulation of Bcl-2 expression and the upregulation of Bcl-2-asscoaited X protein (Bax) expression in BPH tissues were closely associated with such pro-apoptotic activity [89]. 

A study using a rat model with testosterone propionate-induced BPH also showed that the oral administration of anthocyanin (100 mg/kg) led to a significant decrease in prostate weight (*p* < 0.01) ([89]. In this study, in addition to apoptosis, cell proliferation and cell cycle regulation were reported to be related to the anti-growth effects of anthocyanin in a rat BPH model [89]. Briefly, anthocyanin downregulated the expression of proliferating cell nuclear antigen (PCNA), which is a marker of cell proliferation, and cyclin D1, which is a representative regulator of the cell cycle, in BPH tissues [89]. Interestingly, anthocyanin intake decreased androgen metabolism-related factors, such as androgen receptor (AR), 5-α reductase type 2 (5AR2), steroid receptor coactivator 1 (SRC1), and prostate specific antigen (PSA) expression in BPH tissues and serum levels of DHT. Therefore, it is important to understand the molecular mechanisms underlying anthocyanin anti-growth activities in BPH tissues. 

In addition to the rat BPH model, there was a report that anthocyanins extracted from bilberry exerted anti-growth effects on prostatic tissues in testosterone propionate-induced BPH rats [90]. However, we should note that this study showed an additive effect of anthocyanins with the pollen of *Brassica napus L*., which reportedly exerts anti-cancer effects in prostate cancer cells [91]. Furthermore, this study also showed that the regulation of oxidative stress was associated with the anti-growth effects of anthocyanins with pollen of *Brassica napus L*. 

Seoritae, a type of black soybean, and its extract significantly decreased prostate weight via the regulation of oxidative stress, apoptosis, and 5-α reductase activity in BPH rats [92]. A Seoritae extract includes isoflavones and anthocyanins, and authors have hypothesized that part of the pharmacological effects of a Seoritae extract in BPH was caused by 5-α reductase inhibitory effects, as well as the antioxidant effects of the isoflavones and anthocyanin [92]. In the flavanols section above (4.1.1), we introduced the possibility of *Tropaeolum tuberosum* extracts, containing flavonoids, as therapeutic agents. This type of extract includes flavonoids and anthocyanins [62]. Based on these facts, anthocyanin is suggested to be effective as a potential natural therapeutic agent for BPH treatment [88,90,92] and, moreover, they suggest an enhanced necessity of further clinical trials regarding clinical effects and safety in humans.

### 4.2. Non-Flavonoids

#### 4.2.1. Green Tea Polyphenol

Green tea is one of the most popular beverages in the world. Epigallocatechin-3-gallate (EGCG) is the major polyphenol in green tea [93]. Green tea contains various polyphenols and has been shown to be effective in the prevention and treatment of malignant tumors, cardiovascular diseases, and infectious diseases [94].

Several in vivo and in vitro studies have reported the pharmacological effects of green tea polyphenol (especially EGCG) in BPH. For example, in a human benign prostate hyperplasia cell line (BPH-1), EGCG suppressed cell viability in a dose- and time-dependent manner (1–100 μM and 24–48 h) [95]. In addition, this study showed that BPH-1 cell migration was significantly inhibited by 10–100 μM EGCG in a wound healing assay [95]. The anti-proliferative effects of green tea polyphenol were expected according to previous reports on malignancies, including prostate cancer [96,97,98]. However, EGCG can also modulate BPH-1 cell migration, although BPH is a benign disease. Regarding molecular mechanisms, anti-migration effects, changes of the distribution of paxillin, F-actin, and stress fiber were detected via an in vitro study [95]. In addition, this study demonstrated that EGCG treatment significantly decreased the phosphorylation of focal adhesion kinase (FAK) and reduced the protein levels of Ras homolog (Rho) and cell division cycle 42 (Cdc42) in a dose-dependent manner [95]. These focal adhesion-related molecules and regulators of cell shape are associated with cell migration under physiological and pathological conditions [99,100,101]. Finally, EGCG was speculated to play a crucial role in BPH-1 cell migration inhibition due to its flattened and angular morphology. These findings are important for understanding the pathological characteristics of BPH. 

Several investigators have reported the anti-growth effects of ECGC in animal experiments. First, the pharmacological effects of orally-administered EGCG (50 or 100 mg/kg for four weeks) were investigated in rats fed a high-fat diet (HFD) and subcutaneously injected with 10 mg/Kg testosterone (high-fat diet (HDF)-BPH model) [102]. The serum levels of glucose, triglycerides, total cholesterol, and prostate weight in the EGCG-treated HDF-BPH model were significantly lower than those in the untreated HDF-BPH model. In addition, EGCG treatment prevented the change in prostatic morphological structure to that of BPH. Furthermore, EGCG treatment inhibited oxidative stress, as evaluated by SOD, GPx, and MDA levels, and inflammation-related factors, including IL-1β, IL-6, TNF-α, and vascular endothelial growth factor (VEGF), in HDF-BPH rats. Importantly, EGCG-induced antioxidative and anti-inflammatory effects were found in both systemic and prostatic locations [102]. Analyses of prostatic tissue showed that the expression of insulin-like growth factor (IGF)-1 and IGF-2 in EGCG-treated HDF-BPH rats were significantly lower than those in untreated rats, and conversely, the tissue levels of IGF-binding protein-3 (IGFBP-3), peroxisome proliferator-activated receptor (PPAR)-α, and PPAR-γ in HDF-BPH rats were increased by EGCG treatment [102]. Presently, there is a general agreement that the IGF axis, composed of IGF-1, IGF-2, and IGFBP-3, plays important roles in prostate size in BPH [103,104]. Similarly, several previous reports have shown that PPAR-α and -γ are associated with BPH etiology in a rat BPH model [105,106]. Thus, EGCG is speculated to regulate the pathological characteristics of BPH through complex mechanisms. However, we should note that such findings were obtained by animal experiments using rats with metabolic syndrome. 

In a testosterone-induced BPH rat model, the daily intragastric administration of 50 or 100 mg/kg EGCG for four weeks suppressed morphological changes in BPH, such as prostate epithelial cell expansion, papillary protuberance, and collagen deposition [107]. In addition, changes to various oxidative stress-related factors (SOD, GPx, CAT, GSH, total sulfhydryl (TSH), and MDA) and inflammation-related factors (IL-1β, IL-6, and TNF-α) by EGCG were evaluated using various methods [107]. As a result, EGCG increased the prostatic activities of antioxidant enzymes (SOD, GPx, and CAT) and the levels of non-enzymatic antioxidants (GSH and TSH), and it reduced the level of MDA, which is recognized as a lipid peroxidation marker, in BPH rats [107]. This study also showed that EGCG decreased the prostatic levels of pro-inflammatory cytokines and enzymes, including IL-1β, IL-6, TNF-α, and COX-2, in prostatic tissues [107]. In addition to oxidative stress and inflammation, in rats with BPH, EGCG mediated prostatic angiogenesis and epithelial–mesenchymal transition (EMT)-related factors, sex hormone mediators, and a variety of micro (mi) RNAs [107]. In short, EGCG reduced the prostatic expression of VEGF, basic fibroblast growth factor (bFGF), and EGF; decreased the expression of TGF-β1, TGF-β1 receptor 1, and hypoxia-inducible factor-1α (HIF-1α); decreased the phosphorylation-Smad3 (known as mothers against decapentaplegic homolog 3); reduced the expression of AR and ER-α, enhanced ER-β expression; and enhanced miR-133a/b in BPH rats [107]. In previous reports, angiogenesis and EMT were reported to play important roles in the etiology and development of BPH [108,109,110,111]. Unfortunately, there is little information on the pathological roles of miR-133a/b in BPH. However, several reports showed that miR-133a/b was associated with ameliorative effects of prostatic deficits by cadmium and EMT regulation under pathological conditions [112,113]. In addition to such in vivo studies, EGCG was also reported to have anti-inflammatory (down-regulation of IL-1β, IL-16, and TNF-α), anti-growth (down-regulation of IGF-1 and IGF-2), and pro-apoptotic (up-regulation of PPAR-αand -γ) activities by in vitro studies [18]. These findings support the opinion that ECGC acts as a suppressor of BPH via the regulation of complex mechanisms, including angiogenesis and EMT. Summaries of the molecular changes by EGCG are shown in Table 1, Table 2 and Table 3.

Based on these facts, EGCG has been suggested as a potential preventive and therapeutic agent for BPH [95,102,107,115]. Unfortunately, there has been no well-designed clinical trial with a large population to clarify the clinical usefulness and safety in humans. In addition, from the analyzed EGCG functions, there is no information on the pharmacological activity of other types of green tea polyphenols, such as (−)-epicatechin (EC), (−)-epigallocatechin (EGC), and (−)-epicatechin-3-gallate (ECG). The therapeutic and pharmacological effects of green tea polyphenols differ among EGCG, EC, EGC, and ECG under pathological conditions, including prostate cancer [116,117,118]. Therefore, in vitro and in vivo studies employ cell lines and animal models of BPH, respectively, to determine the biological roles of EC, EGC, and ECG, which are also essential to discuss prevention and treatment strategies of BPH. 

#### 4.2.2. Resveratrol

Resveratrol is a polyphenol found in grapes, red wine, peanuts, chocolate, cocoa, and berries [119]. Its structure is shown in Figure 5. 

Resveratrol has been reported to have anti-proliferative effects not only in cancer cells but also in non-tumor cells, such as pulmonary artery smooth muscle cells [120]. In an in vitro study, resveratrol inhibited the growth of BPH-1 cells [121]. In short, flow cytometry indicated that apoptotic cell rates significantly increased in a concentration-dependent manner when resveratrol was administered to BPH-1 cells at various concentrations (20 and 30 μM). Furthermore, a Western blot analysis indicated that phosphorylated-p38 mitogen-activated protein kinase (MAPK) level significantly increased after resveratrol treatment, and the expression of Forkhead-box protein O3a (FOXO3a) and anti-apoptotic molecules, such as Bcl2 and Bcl-XL, were downregulated with increased cleaved caspase3 levels. Finally, resveratrol treatment led to enhanced cleaved caspase3 levels [121]. These results suggested that resveratrol may induce apoptosis via p38 MAPK activation and FOXO3a repression. Furthermore, this study showed that when BPH-1 cells were co-treated with a p38 MAPK inhibitor, SB203580, or an ROS scavenger, N-acetyl-L-cysteine (NAC), the levels of SOD2 and CAT (which are recognized as antioxidative enzymes) and Bcl2 and Bcl-XL (which are anti-apoptotic markers) were elevated. Furthermore, cleaved caspase3 levels were decreased by such additional treatments [121]. Based on these findings, the authors suggested that resveratrol activated p38 MAPK and suppressed FOXO3a, thereby suppressing antioxidative activity and increasing ROS accumulation, leading to the apoptosis of BPH-1 cells [121]. 

To clarify the in vivo biological activities of resveratrol, its pro-apoptotic function was investigated using an animal model of BPH [122]. In this study, resveratrol was administered daily for four weeks at a dose of 1 mL per 100 g of body weight. Resveratrol treatment led to a decrease in prostate weight. In addition, a Western blot analysis of prostate tissues revealed a decrease in iNOS and COX-2 protein expression in the resveratrol-treated group. These results indicate that resveratrol exerts anti-inflammatory effects in BPH tissues [122]. In addition, resveratrol decreased the expression of anti-apoptotic molecules Bcl-2 and Bcl-XL, and it increased the expression of the pro-apoptotic protein Bax. These findings support the opinion that resveratrol plays an anti-proliferative role through the regulation of inflammation and apoptosis in BPH [122]. 

Another study investigated the anti-growth effects of resveratrol using an obesity-induced mouse BPH model [119]. In this study, mice were treated with resveratrol (100 mg/kg once daily by oral gavage) from the 10th to 12th week of diet. At first, when ROS levels were measured in prostate tissues via the in situ quantification of dihydroethidium fluorescence, the fluorescent intensity was 98% higher in obese mice. However, resveratrol treatment fully restored the increased ROS production and the overexpression of glycosylated 91-kDa glycoprotein (gp91phox) and nerve growth factor (NGF) mRNA. Moreover, resveratrol increased insulin-induced phosphor-AKT expression in obese mice, although the impairment of insulin signaling was evidenced by low pAKT expression. These results indicate the possibility that resveratrol treatment is a valuable and useful alternative prevention strategy for obesity-related BPH [119]. 

In contrast to these reports, there was an opinion that the anti-growth effect of resveratrol by resveratrol administration was minimal without significant levels [123]. This clinical study was performed in a randomized, placebo-controlled manner using resveratrol (150 or 1000 mg) for four months. Prostate size and serum levels of PSA and sex steroid hormones were measured in 66 men with metabolic syndrome. As a result, high-dose resveratrol (1000 mg daily) significantly reduced prostate size and serum levels of dehydroepiandrosterone (DHEA) and DHEA sulfate (DHEAS), while circulating levels of PSA, testosterone, free testosterone, and DHT did not change [123]. Finally, the authors concluded that resveratrol administration had no significant effect on the suppression of prostate volume. We agree with their opinion derived from these results; however, the limitations of these clinical trials should be noted. In short, the duration of resveratrol intake was only four months. In addition, although 150–1000 mg/dy/day resveratrol were administered in this clinical trial, 20–100 mg/Kg/day were administered in previous animal experiments that showed significant effects of resveratrol [119,124]. Thus, there is a possibility that the duration and dose of resveratrol are insufficient to show anti-proliferative effects in humans.

#### 4.2.3. Cacao Polyphenol

ACTICOA^TM^ (Barry Callebaut France, Louviers, France) powder (AP) is a cocoa polyphenol extract. Research indicates that oral treatment with AP prevents BPH [125]. AP was administered to rats orally at doses of 24 and 48 mg/kg/day for two weeks. Thereafter, serum dihydrotestosterone was measured and the prostate was removed and weighed to determine efficacy. The results showed that serum DHT levels and the prostate size ratio (prostate weight/rat body weight) were significantly reduced in a dose-dependent manner, indicating that the treatment was effective in reducing BPH [125]. 

### 4.3. Metabolites from Flavonoids

#### 4.3.1. Protocatechuic Acid

Protocatechuic acid (PA: 3,4-dihydroxybenzoic acid) is a major metabolite of anthocyanins, and it has been found in a large variety of vegetables and fruits [126,127]. Its structure is shown in Figure 6.

It has been recognized as an effective antioxidant and anti-inflammatory factor. In an in vivo study using a testosterone-induced BPH rat model, the administration of PA (40 mg/kg) for four weeks decreased prostatic weight by 19% and decreased the serum and prostate levels of myeloperoxidase activities and nitrate oxide, which are reliable markers of the degree of inflammation and oxidative stress [128]. In addition, PA reduced the serum levels of proinflammatory cytokines, including IL-1β and TNF-α, in BPH rats. Thus, inflammation and oxidative stress markers in BPH rats were attenuated upon treatment with PA. These results have shown that PA prevents the progression of BPH in rats through its antioxidant and anti-inflammatory mechanisms [128].

To our knowledge, there have been no in vivo studies reporting the direct activity of PA on BPH, except for the above-mentioned report [128]. However, a study reported that the natural compound, including PA, improved the pathological features of BPH, such as prostate enlargement and smooth muscle layer thickness in rats [129]. Specifically, *Cynomorium songaricum Rupr* (CS) belongs to the genus of parasitic perennial flowering plants, is mostly used as a traditional medicine, and includes various natural extracts, such as luteolin, gallic acid (GA), and PA. This study also showed that PA significantly inhibited the DHT- or 17β-estradiol-induced proliferation of BPH-1 cells [129]. Thus, PA is speculated to exert anti-growth effects on BPH in vivo and in vitro. However, further studies are necessary to conclude the clinical usefulness and detailed molecular mechanisms of BPH.

#### 4.3.2. Equol

Equol is a polyphenol/isoflavonoid molecule produced from the soybean isoflavones daidzin and daidzein by gut bacteria [130,131]. Its structure is shown in Figure 7. 

Equol specifically binds to 5α-DHT, and it reduces rat prostate size, serum 5α-DHT levels, and the action of androgen hormones [132]. A study investigating the effects of low-dose equol supplements (6 mg, with a meal twice daily) for four weeks in patients with moderate or severe BPH showed an improvement in IPSS scores [133]. In addition, patients with severe BPH display a 21% reduction in 5α-DHT levels [133]. Equol has powerful antioxidant and anti-aging properties that reduce prostate inflammation and latent neoplastic growth [134]. This mechanism is similar to that mentioned above, and equol specifically binds to 5α-DHT by sequestering 5α-DHT from the androgen receptor [134].

## 5. Conclusions

This review summarizes the potential preventive effects of polyphenol intake on BPH by presenting the thus far published data obtained from in vitro studies, animal studies, and clinical trials. In vitro studies using BPH cell lines have shown the anti-inflammatory and antioxidant effects of polyphenols, and several in vivo studies using animal models have shown similar results. The treatment of BPH includes conservative treatment with oral medications and surgical treatment, such as transurethral resection. However, each treatment has its own side effects and complications that should be carefully considered. Treatment with phytochemicals, such as polyphenols as reviewed here, may solve this problem. Though the anti-BPH effect of polyphenols alone is believed to be limited, a combination of conventional treatments with polyphenols may provide some clinical benefit. Therefore, further research is required on the effect of polyphenols on BPH.

## Figures and Tables

**Figure 1 molecules-26-00450-f001:**
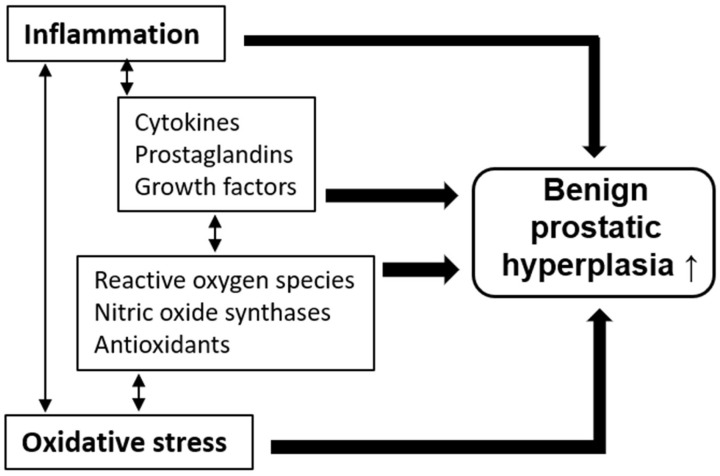
Pathological mechanisms of benign prostatic hyperplasia (BPH) development via inflammation and oxidative stress.

**Figure 2 molecules-26-00450-f002:**
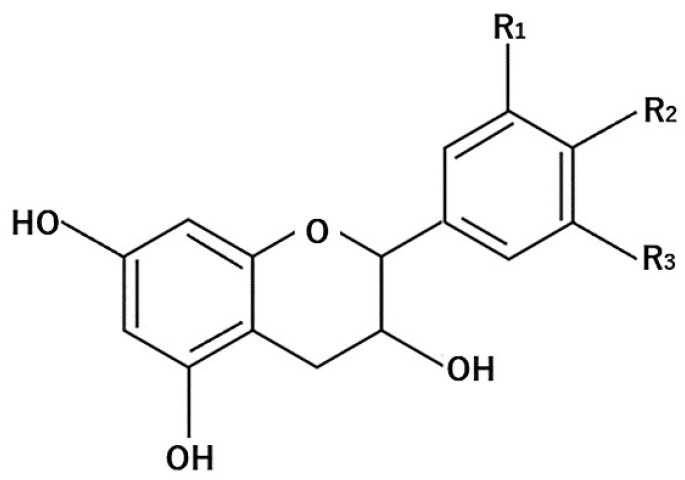
General structure of flavanols.

**Figure 3 molecules-26-00450-f003:**
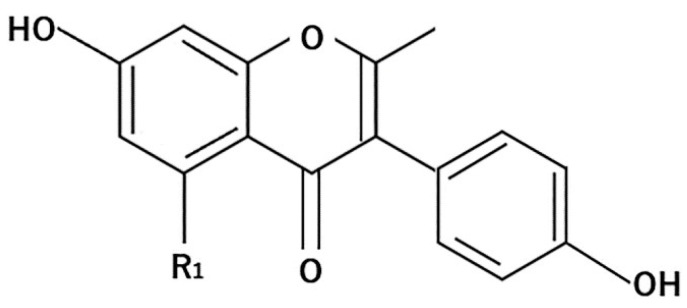
General structure of isoflavones.

**Figure 4 molecules-26-00450-f004:**
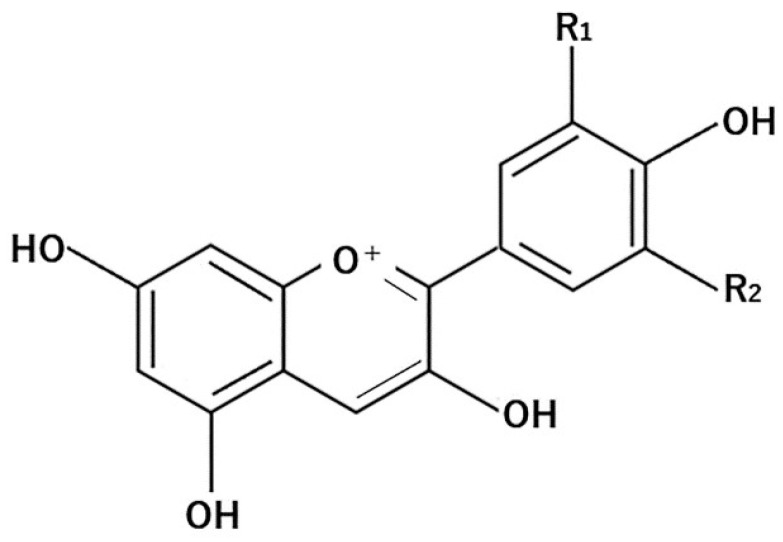
General structure of anthocyanins.

**Figure 5 molecules-26-00450-f005:**
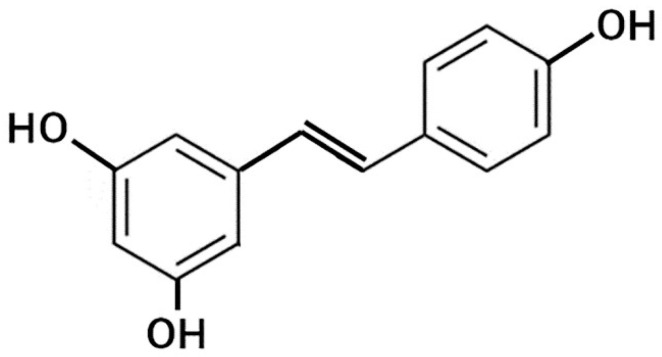
General structure of resveratrol.

**Figure 6 molecules-26-00450-f006:**
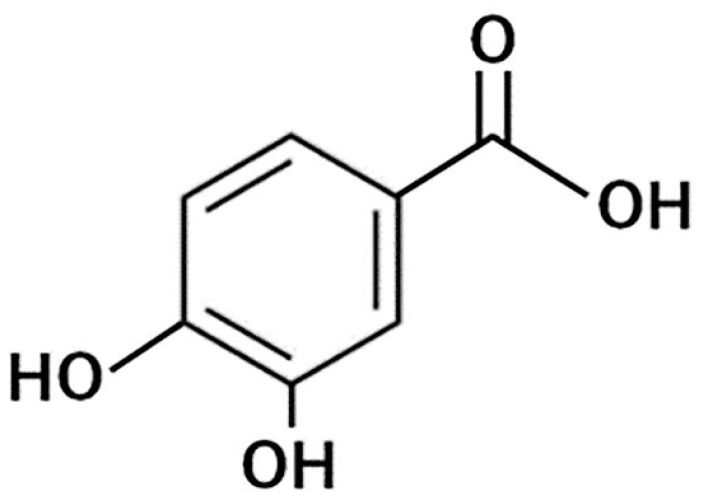
General structure of protocatechuic acid.

**Figure 7 molecules-26-00450-f007:**
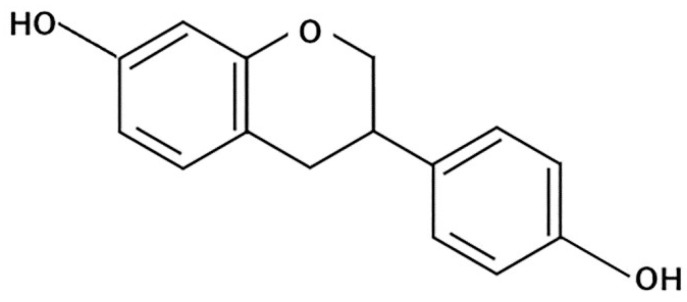
General structure of equol.

**Table 1 molecules-26-00450-t001:** Change of oxidative stress- and inflammatory-related molecules by green tea polyphenol.

Molecules	Sample	Change	Author/Year/Reference
Anti-oxidants			
CAT	Tissue	↑	Zhou, J./2018/[107]
GPX	Blood	↑	Chen, J./2016/[102]
	Tissue	↑	Chen, J./2016/[102]
	Tissue	↑	Zhou, J./2018/[107]
GSH	Tissue	↑	Zhou, J./2018/[107]
SOD	Blood	↑	Chen, J./2016/[102]
	Tissue	↑	Chen, J./2016/[102]
	Tissue	↑	Zhou, J./2018/[107]
TSH	Tissue	↑	Zhou, J./2018/[107]
Oxidative marker			
MDA	Blood	↓	Chen, J./2016/[102]
	Tissue	↓	Chen, J./2016/[102]
	Tissue	↓	Zhou, J./2018/[107]
Pro-inflammation			
COX-2	Tissue	↓	Zhou, J./2018/[107]
IL-1β	Blood	↓	Chen, J./2016/[102]
	Tissue	↓	Chen, J./2016/[102]
	Tissue	↓	Zhou, J./2018/[107]
	Cell line	↓	Cicero, A.F.G./2019/[18]
IL-6	Blood	↓	Chen, J./2016/[102]
	Tissue	↓	Chen, J./2016/[102]
	Tissue	↓	Zhou, J./2018/[107]
IL-16	Cell line	↓	Cicero, A.F.G./2019/[18]
P65; total	Tissue	→	Zhou, J./2018/[107]
P65; phosphorylated	Tissues	↓	Zhou, J./2018/[107]
TNF-α	Blood	↓	Chen, J./2016/[102]
	Tissue	↓	Chen, J./2016/[102]
	Tissue	↓	Zhou, J./2018/[107]
	Cell line	↓	Cicero, A.F.G./2019/[18]

CAT, catalase; GPX, glutathione peroxidase; GSH, glutathione; SOD, superoxide dismutase; TSH, total sulfhydryl; MDA, malondialdehyde; COX-2, cyclooxygenase-2; IL-1β, interleukin-1β; IL-6, interleukin-6; TNF-α, tumor necrosis factor-α.

**Table 2 molecules-26-00450-t002:** Changes of hormone-related molecules and growth factors by green tea polyphenol.

Molecules	Sample	Change	Author/Year/Reference
Hormonal			
5α-reductase	Tissues	↓	Liao, S./2001/[114]
AR	Tissues	↓	Zhou, J./2018/[107]
ER-α	Tissues	↓	Zhou, J./2018/[107]
ER-β	Tissues	↑	Zhou, J./2018/[107]
Growth factors			
Basic FGF	Tissue	↓	Zhou, J./2018/[107]
EGF	Tissue	↓	Zhou, J./2018/[107]
IGF-1	Tissue	↓	Chen, J./2016/[102]
	Cell line	↓	Cicero, A.F.G./2019/[18]
IGF-2	Tissue	↓	Chen, J./2016/[102]
	Cell line	↓	Cicero, AFG/2019/[18]
IGFBP-3	Tissue	↑	Chen, J./2016/[102]
TGF-β1	Tissue	↓	Zhou, J./2018/[107]
TGF-β1 receptor	Tissue	↓	Zhou, J./2018/[107]
VEGF	Blood	↓	Chen, J./2016/[102]
	Tissue	↓	Chen, J./2016/[102]
	Tissue	↓	Zhou, J./2018/[107]

AR, androgen receptor; ER-α, estrogen receptor-α; ER-β, estrogen receptor-β; FGF, fibroblast growth factor; EGF, epidermal growth factor; IGF, insulin-like growth factor; IGFBP-3, insulin-like growth factor binding protein-3; TGF-β1, transforming growth factor-β1; VEGF, vascular endothelial growth factor.

**Table 3 molecules-26-00450-t003:** Changes of biological factors by green tea polyphenol.

Molecules	Sample	Change	Author/Year/Reference
α-SMA	Tissue	↓	Zhou, J./2018/[107]
Cdc42	Cell line	↓	Tepedelen, B.E./2017/[95]
E-cadherin	Tissue	↑	Zhou, J./2018/[107]
FAK; total	Cell line	→	Tepedelen, B.E./2017/[95]
FAK; phosphorylated	Cell line	↓	Tepedelen, B.E./2017/[95]
Fibronectin	Tissue	↓	Zhou, J./2018/[107]
HIF-1α	Tissue	↓	Zhou, J./2018/[107]
miR-133a	Tissue	↑	Zhou, J./2018/[107]
miR-133b	Tissue	↑	Zhou, J./2018/[107]
PAK	Cell line	↓	Tepedelen, B.E./2017/[95]
PPAR-α	Tissue	↑	Chen, J./2016/[102]
	Cell line	↓	Cicero, A.F.G./2019/[18]
PPAR-γ	Tissue	↑	Chen, J./2016/[102]
	Cell line	↓	Cicero, A.F.G./2019/[18]
Paxillin	Cell line	↓	Tepedelen, B.E./2017/[95]
Rho A	Cell line	↓	Tepedelen, B.E./2017/[95]
Smad3; total	Cell line	→	Zhou, J./2018/[107]
Smad3; phosphorylated	Cell line	↓	Zhou, J./2018/[107]

α-SMA, α-smooth muscle actin; Cdc42, cell division cycle 42; FAK, focal adhesion kinase; HIF-1α, hypoxia-inducible factor-1α; PAK, p21 protein-activated kinase; PPAR-α, peroxisome proliferator activated receptor-α; PPAR-γ, peroxisome proliferator activated receptor-γ; Rho A, Ras homolog gene family, member A.

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
