# Peer review of "Pharmacological Effects and Potential Clinical Usefulness of Polyphenols in Benign Prostatic Hyperplasia"

_molecules, 2021, doi:10.3390/molecules26020450_

Round 1

Reviewer 1 Report

Pag 2 line 66-67 Explain this sentence better: These exert a weak estrogenic effect, and are thought to exert an anti-estrogen effect in vivo

70 Put the full stop in : researchers Such foods

71-72 lose the line: Such foods have advantages because they are less toxic, more effective, and economical, and there is growing interest in plant-derived natural antioxidants.

Pag 3 line 140-141 Prostaglandins, an inflammatory mediator, have been observed in the prostate ( healthy or hypertrophic?)

pag 4 156 lose the line: the most abundant antioxidants in the human diet

Figure 2. correct : Isofulavones.

The scientific names of the plants must be written in italics, the name of the author must be put always or never( Abacopteris penangiana - Tropaeolum tuberosum -Pueraria mirifica-----Brassica napus L. - Cynomorium songaricum Rupr (CS))

As for formulas of chemical compounds, or make them of all or none (flavonol/isoflavones/anthocyanins yes and resveratrolo/protocatechuic acid/equol not ?)

Author Response

Dear reviewer:

We thank you for carefully evaluating our manuscript and are happy to receive your positive evaluation. In the revised version of the manuscript, we have modified some sentences, tables, and added new figures in response to the other reviewers’ suggestions. Changes made in response to reviewers’ comments are highlighted in red in the revised version of the manuscript.

Comment 1) Pag2 2 line 66-67 Explain this sentence better: These exert a weak estrogenic effect, and are thought to exert an anti-estrogen effect in vivo. 

Response:  

We thank you for this suggestion. We too felt that we had drafted the sentence with an unclear meaning. Therefore, we have rephrased the sentence as follows, in order to make it easier for the readers to understand: “These exert some degree of estrogenic effects, and are thought to inhibit prostatic cell growth” (Page 2, line 66-67).

(Comment 2) line 70 Put the full stop in : researchers Such foods / 71-72 lose the line: Such foods have advantages because they are less toxic, more effective, and economical, and there is growing interest in plant-derived natural antioxidants.

Response:

Thank you for the suggestions. We are sorry for the typo. We have also modified the sentence as follows: “In recent years, phytochemicals found in fruits, vegetables, and tea, have gained the interest of researchers because they are less toxic, more effective, and economical.” (Page 2, line 69-71). With this modification based on your suggestion, we feel that these sentences convey what we want to say in the introduction section.

(Comment 3) Page 3 line 140-141 Prostaglandins, an inflammatory mediator, have been observed in the prostate (healthy or hypertrophic?)

Response:

Thank you for the important question. In the study by Altavilla D. et al, [39], increased levels of prostaglandins were detected in BPH. Therefore, we modified this sentence according to the results of the study (Page 3, line 141).

(Comment 4) page 4 156 lose the line: the most abundant antioxidants in the human diet

Response:

We agree with your opinion. In the revised version of the manuscript, this sentence was modified as follows: “Polyphenols are the most abundant antioxidants in many foods, such as fruits, vegetables, seeds, nuts, chocolate, wine, coffee, and tea” (Page 4, line 165-167).

(Comment 5) Figure 2. correct : Isofulavones.

Response:  

We are sorry for the simple mistake. We have modified the title of Figure 3.

(Comments 6) The scientific names of the plants must be written in italics, the name of the author must be put always or never (Abacopteris penangiana - Tropaeolum tuberosum -Pueraria mirifica-----Brassica napus L. - Cynomorium songaricum Rupr (CS)).

Response:  

Thank you for this important suggestion. We have checked and modified them at all instances in the text.

(Comment 7) As for formulas of chemical compounds, or make them of all or none (flavonol/isoflavones/anthocyanins yes and resveratrolo/protocatechuic acid/equol not ?)

Response:  

   We had initially planned to show the structures of all compounds. However, we were afraid that the main purpose of this review would become unclear if we added in all the structures. In addition, these information are clearly provided in previous excellent reviews. On the other hand, your question is important for readers too. Therefore, we added the information referring to previous reviews in Page 2, line 72-74.

Reviewer 2 Report

Reviewer comments and suggestions

Reviewer comments and suggestions

The current review introduced the relationships among oxidative stress, chronic inflammation, and/or angiogenesis, in the BPH and polyphenols treatment in disease prevention. Furthermore, the study discusses in vivo and in vitro pharmacological effects and molecular mechanisms of polyphenols against BPH.

The representation of the paper was not good. The authors need to draw two figures that should be focused on the mechanism and tables need to add more data in that. Besides the above, I am suggesting some major comments to be incorporated in the revised version of the MS.

Below are the comments that need to update in the revised version of the manuscript.

  1. Line 23-27, it’s not needed to include big sentence in the abstract.
  2. Line 39, pls explain with the clinical manifestation of BPH here only
  3. Line 57, Seems incompleted sentence (13)
  4. Line 59-60, it is better to explain which infection, please explore it
  5. Line 67, These substances “These pharmacological products”
  6. Section 84, It is better to include at least one figure that shows the mechanism that the author has discussed in the paper and try to point out some polyphenol that already used in the treatment. This review is like a review of the literature with no sufficient background of clinical information related to BPH
  7. Table 1 PLEASE SPECIFY THE TABLE THAT THE AUTHOR USED THIS TREATMENT FOR various links with BPH. I am not convinced with the table
  8. Table 2 I think only two papers the author explored, should be rewritten with other published studies
  9. Line 545-546 conclusion should be written in a way that consists of only important points rather than discussing other things
  10. Line 551, did the author think that only polyphenol can do. other products may not have the potential
  11. It should be present that why oral medication may provide some clinical benefit (discussion needed)
  12. Please check the reference number, 28,42, 49 and 50,53

Author Response

Reviewer comments and suggestions

The current review introduced the relationships among oxidative stress, chronic inflammation, and/or angiogenesis, in the BPH and polyphenols treatment in disease prevention. Furthermore, the study discusses in vivo and in vitro pharmacological effects and molecular mechanisms of polyphenols against BPH.

The representation of the paper was not good. The authors need to draw two figures that should be focused on the mechanism and tables need to add more data in that. Besides the above, I am suggesting some major comments to be incorporated in the revised version of the MS.

Below are the comments that need to update in the revised version of the manuscript.

Response:

   We thank you for carefully evaluating our manuscript. We agree with your opinions and suggestions. Therefore, in revised version of the manuscript, we have modified some sentences, tables, and added new figures according to your suggestions. We believe that contents of our review have improved with these modifications. Changes made in response to the reviewers’ comments are highlighted in red in the revised version of the manuscript.

  1. Line 23-27, it’s not needed to include big sentence in the abstract.

Response:

We thank you for this suggestion and agree with your opinion. Therefore, this sentence was divided into 3 smaller sentences based on the contents (Page 1, line 23-27). We believe that the purport of the abstract in revised manuscript still remains similar to that in the original one, even after the changes.

  1. Line 39, pls explain with the clinical manifestation of BPH here only

Response:

Thank you for your suggestion. As suggested, this sentence about the physical appearance of the prostrate is not a good fit for this type of review. Additionally, we also feel that this sentence is not necessary in this section and hence have deleted this sentence in the revised version of the manuscript.

  1. Line 57, Seems incompleted sentence (13)

Response:

We are sorry for the simple mistake. We have completed this sentence by adding a citation about an example of the cytokines, which we are referring to (Page 2, line 56-57).

  1. Line 59-60, it is better to explain which infection, please explore it 

Response:

Thank you for your important suggestion. In the previous report being cited here [reference 14], the authors paid special attention to inflammation and immune response caused by bacteria or other foreign antigens in BPH. Therefore, based on this study and your suggestion, we have added more information regarding the cause of infection in this sentence (Page 2, line 59).

  1. Line 67, These substances “These pharmacological products”

Response:

According to your suggestion, we have modified the sentence (Page 2, line 67). Thank you for your advice.

  1. Section 84, It is better to include at least one figure that shows the mechanism that the author has discussed in the paper and try to point out some polyphenol that already used in the treatment. This review is like a review of the literature with no sufficient background of clinical information related to BPH 

Response:

Thank you very much for this important suggestion. We agree with your opinion. We added the schema of pathological mechanisms of BPH via regulation of inflammation and oxidative stress as a new figure, Figure 1: Pathological mechanisms of BPH development via inflammation and oxidative stress. In addition, we also added the comments about it in the section 2.3. Benign prostate hyperplasia and oxidative stress (Page 4, line 154-158).

  1. Table 1 PLEASE SPECIFY THE TABLE THAT THE AUTHOR USED THIS TREATMENT FOR various links with BPH. I am not convinced with the table.

and

  1. Table 2 I think only two papers the author explored, should be rewritten with other published studies 

Response:

Thank you for your important suggestions. We have re-searched the literature for other published studies about the relationship between green tea polyphenol and molecular mechanisms of BPH growth and we found two studies, which have mentioned  about this relation [reference 18 and newly added reference 117]. In the revised version of our manuscript, we added these results into the section 4.2.1 Green tea polyphenol (Page 10, line 424-427) and also in Table 1 – 3

  1. Line 545-546 conclusion should be written in a way that consists of only important points rather than discussing other things

Response:

We thank you for the suggestion and agree with your opinion. This sentence was deleted from the conclusions section.

  1. Line 551, did the author think that only polyphenol can do. other products may not have the potential

and

  1. It should be present that why oral medication may provide some clinical benefit (discussion needed) 

Response:

In this sentence, we wanted to emphasize that a combination of conventional therapies and polyphenols may be effective in BPH treatment. Therefore, we have modified this sentence in the revised version of the manuscript (Page 14, line 562-563). 

  1. Please check the reference number, 28, 42, 49 and 50, 53 

Response:

We are sorry for the mistakes. We have cross-checked all references including 28,42, 49, 50, and 53, and modified the wrong references.

Reviewer 3 Report

This review by Mitsunari et al. summarizes the research progress of polyphenols including their pharmacological effects and clinical applications. The topic is generally interesting, and the summarized progress is useful in the field of drug discovery. The manuscript is also very well-written. The only suggestion I have is that as a review article, it would be nice to include one or more pictures describing the mechanism-of-action of the compounds mentioned. This may help improve the impact of this review.

Author Response

This review by Mitsunari et al. summarizes the research progress of polyphenols including their pharmacological effects and clinical applications. The topic is generally interesting, and the summarized progress is useful in the field of drug discovery. The manuscript is also very well-written. The only suggestion I have is that as a review article, it would be nice to include one or more pictures describing the mechanism-of-action of the compounds mentioned. This may help improve the impact of this review.

Response:

We thank the reviewers for carefully evaluating our manuscript. We are happy to receive your positive evaluation. In the revised version of the manuscript, we modified some sentences and all tables, and also added a new Figure (Figure 1: Pathological mechanisms of BPH development via inflammation and oxidative stress) according to your and the other reviewers’ suggestions. Changes made in response to the reviewers’ comments are highlighted in red in the revised version of the manuscript.

Round 2

Reviewer 2 Report

Please check Non-flavonoids groups, are these natural products are really non-flavonoids? Please check it.

Author Response

Reviewer comments

Please check Non-flavonoids groups, are these natural products are really non-flavonoids? Please check it

Response:

We thank you for carefully evaluating our manuscript. As you pointed out, protocatechuic acid and equol were metabolites from flavonoids (anthocyanins and isoflavones). In that sense, they can be classified as members of flavonoids. However, we want to show them in independent section from “4.1. Flavonoids”. Therefore, we added new section “4.3. Metabolites from flavonoids” section (page 13: line 521), and protocatechuic acid and equol were showed in this new section of the revised manuscript. In addition, we added the comments about these issues (page 13: lines 523 – 524 and page 13: lines 546 – 547) and cited new 2 references (new reference [131] and [135]) to clarify the fact that they were metabolites from the flavonoids. We believe that such modification leads to avoid the misunderstanding about protocatechuic acid and equol in this review.

We appreciate for your very important opinion
